



# Lower tropospheric ozone over India and its linkage to the South Asian monsoon

Xiao Lu[1,2], Lin Zhang[1], Xiong Liu[3], Meng Gao[2], Yuanhong Zhao[1], Jingyuan Shao[1]

5      (1) Laboratory for Climate and Ocean-Atmosphere Studies, Department of Atmospheric and Oceanic

Sciences, School of Physics, Peking University, Beijing 100871, China

(2) School of Engineering and Applied Sciences, Harvard University, Cambridge, MA, 02138, USA

(3) Harvard-Smithsonian Center for Astrophysics, Cambridge, MA, 02138, USA

*Correspondence to:* Lin Zhang (zhanglg@pku.edu.cn**)**



## Abstract

Lower tropospheric (surface to 600 hPa) ozone over India poses serious risks to local human and crops, and potentially affects global ozone distribution through frequent deep convection in tropical regions. Our current understanding of processes controlling seasonal to long-term variations in lower tropospheric ozone over this region is rather limited due to spatially and temporally sparse observations. Here we present an integrated process analysis of the seasonal cycle, interannual variability, and long-term trends of lower tropospheric ozone over India and its linkage to the South Asian Monsoon using the Ozone Monitoring Instrument (OMI) satellite observations for years 2006–2014 interpreted with a global chemical transport model (GEOS-Chem) simulation for 1990–2010. OMI observed lower tropospheric ozone over India averaged for 2006–2010 show the highest concentrations (54.1 ppbv) in the pre-summer monsoon season (May) and the lowest concentrations (40.5 ppbv) in the summer monsoon season (August). Process analyses in GEOS-Chem show that hot and dry meteorological conditions and active biomass burning together contribute to 5.8 Tg more ozone produced in the lower troposphere of India in May than January. The onset of the summer monsoon brings ozone-unfavorable meteorological conditions and strong upward transport, all lead to large decreases in the lower tropospheric ozone burden. Interannually, we find that both OMI and GEOS-Chem indicate strong interannual positive correlations ($r = 0.55$–$0.58$) between ozone and surface temperature in pre-summer monsoon seasons, with larger correlations found in high $NO_x$ emission regions reflecting $NO_x$-limited production conditions. Summer monsoon seasonal mean ozone levels are strongly controlled by monsoon strengths. Lower ozone concentrations are found in stronger monsoon seasons mainly due to less ozone net chemical production. Furthermore, model simulations over 1990–2010 estimate a mean annual trend of $0.19 \pm 0.07$ (p-value < 0.01) ppbv year$^{-1}$ in Indian lower tropospheric ozone over this period, which are mainly driven by increases in anthropogenic emissions with small contribution (about 7%) from global methane concentration increases.



## 1. Introduction

Ozone in the lower troposphere is a harmful air pollutant for both human and ecosystems (Monks et al., 2015), and plays a central role in atmospheric chemistry as the major source of hydroxyl radicals (OH) (Jacob, 2000). It is also a short-lived greenhouse gas with a global positive radiative forcing of 0.40 (0.20–0.60) W m$^{-2}$ since the preindustrial era (Myhre et al., 2013; Stevenson et al., 2013). Tropospheric ozone is produced by sunlight-driven photochemical oxidation of carbon monoxide (CO) and hydrocarbons in the presence of nitrogen oxide ($NO_x \equiv NO + NO_2$). These ozone precursors are released not only from anthropogenic sources such as industry and transportation, but also from a number of climate-sensitive natural sources such as lightning, biomass burning, and biogenic emissions. It is also transported from the stratosphere (about 550 Tg year$^{-1}$ or 10% of chemical production in the troposphere) (Stohl, 2003; Stevenson et al., 2006). Tropospheric ozone burden (present-day 337 ± 23 Tg) has enhanced 43% since the preindustrial era due to rapid industrialization (Young et al., 2013). Zhang et al. (2016) recently revealed that increases in the tropospheric ozone burden in the past 30 years were dominated by the equatorward redistribution of anthropogenic emissions to developing regions such as East and South Asia, raising increasing interests on ozone pollution over those regions.

Unlike developed regions such as Europe and eastern US, where anthropogenic emission reductions have led to surface ozone levels flatten or decrease since 1990s (Parrish et al., 2012; Cooper et al., 2012; Oltmans et al., 2013; Strode et al., 2015; Lin et al., 2017), developing countries such as China and India have been experiencing anthropogenic emission rises and ozone enhancements (Cooper et al., 2014, Sun et al., 2016; Wang et al., 2017). Recent studies have shown that $NO_x$ emissions in China began to decrease since 2012 due to stringent air pollution controls (Krotkov et al., 2016; Liu et al., 2017). However, air quality in India is continuously deteriorating as indicated by increasing $NO_2$



columns observed from satellite (Krotkov et al., 2016; Geddes et al., 2016), and may become worse in the near future considering projected trends in population and associated anthropogenic emissions (Ghude et al., 2016). Exposure of ozone pollution in India is estimated to cause 12,000 premature deaths in 2011 by chronic obstructive pulmonary disease (Ghude et al., 2016), and up to 36% loss of

wheat and other crop productions in India (Ramanathan et al., 2014; Sinha et al, 2015). In addition, frequent deep convection in tropical Asia allows the uplifted pollutants to influence global ozone distribution (Lelieveld et al., 2001; Sahu et al., 2006; Beig and Brasseur, 2007; Park et al., 2007; Lawrence and Lelieveld, 2010; Srivastava et al., 2012a; Lal et al., 2013). A better understanding of processes controlling lower tropospheric ozone over India thus becomes important to address its local

and global environmental effects.

Distinct seasonal transitions in prevailing wind and rainfall associated with the monsoon circulation result in unique ozone variations in South and East Asia. Winter monsoon prevails in October to March and brings dry and cool weather conditions. With the onset of South Asian (East Asian)

summer monsoon in May, stronger westerlies (southerlies) bring marine air from Arabian Sea (western Pacific) to the Indian subcontinent (East Asia), resulting in significant enhancement of cloud fractions and rainfall (Wang and Lin, 2002; Ding and Chan, 2005). Decreases of tropospheric ozone with the summer monsoon in South and East Asia have been reported from surface (Lal et al., 2000; Naja and Lal, 2002; Naja et al., 2003; Beig et al., 2007; Reddy et al., 2008; Wang et al., 2009; Kumar et al.,

2010; Ding et al., 2013; Hou et al., 2015), ozonesonde (Ojha et al., 2012; Zhou et al., 2013; Lal et al., 2014; Sahu et al., 2014), and satellite observations (Liu et al., 2009; Dufour et al., 2010; Safieddine et al., 2016). A number of modelling studies attribute the ozone minimum in summer time over India to transport of clean marine air (Lal et al., 2014; Sahu et al., 2014) or reduced ozone photochemical production (Roy et al., 2008; Kumar et al., 2012). This seasonality is in contrast to that at mid-latitudes

where surface ozone levels are usually higher in spring and summer due to stronger stratosphere-to-



troposphere transport and photochemistry (Parrish et al., 2013; Cooper et al., 2014).

Most of the above-mentioned studies used individual ground-based observations or regional chemistry models to study seasonal or short-term interannual (up to 5 years) variability of tropospheric ozone in

India. Long-term ground-based ozone observations are extremely scarce in South Asia (Cooper et al., 2014). We also lack a comprehensive analysis of spatiotemporal distribution of lower tropospheric ozone at domestic India scale. In particular, key processes that influence the tropospheric ozone budget over India have not been analyzed and quantified. In this study, we present an integrated analysis of the processes controlling lower tropospheric (surface to 600 hPa) ozone concentrations over India and

their linkage to the South Asian monsoon. Satellite observations from the Ozone Monitoring Instrument (OMI) over 2006–2014 and simulations with the GEOS-Chem chemical transport model (CTM) for 1990-2010 are used to analyze the spatial, seasonal, and interannual variability of lower tropospheric ozone pollution over India before and during the South Asian summer monsoon. We will further examine the potential drivers of long-term trends in lower tropospheric ozone over India.


## 2. Observations and model description

## 2.1. OMI satellite observations

The OMI instrument is onboard the NASA Earth Observing System (EOS) Aura satellite launched in July 2004 with an ascending equator crossing time of ~13:45 LT (local time) (Schoeberl et al., 2006).

OMI is a nadir-viewing instrument that measures backscattered solar radiation in the 0.27–0.5 μm wavelength range with a spectral resolution of 0.42–0.63 nm (Levelt et al., 2006). Its nadir footprint has a spatial resolution of $13\times24$ km$^2$ with near-daily global coverage achieved by a wide view field of 114° and a 2600 km wide swath.

We use the OMI PROFOZ ozone profile retrievals developed by Liu et al. (2010) based on the optimal



estimation method (Rodgers, 2000). Details of the PROFOZ product have been given in Liu et al. (2010) and Kim et al. (2013), and are comprehensively validated recently by Huang et al. (2017a, b). This OMI ozone profile algorithm retrieves partial ozone columns for 24 layers with about 2.5 km thickness for each layer. Validation of the surface–550/750 hPa ozone columns in the tropics with

ozonesonde measurements shows a mean bias within 5% (Huang et al., 2017a). Here we grid the monthly mean OMI data to the 2°×2.5° horizontal resolution with focus on the spatial and temporal distributions of Indian lower tropospheric ozone concentrations for the period of 2006–2014. Comparisons of model simulations with OMI observations need to consider OMI a priori profiles and averaging kernel matrices as described in Zhang et al. (2010). OMI a priori profiles are from the

monthly ozone profile climatology of McPeters et al. (2007). The degrees of freedom for signals for OMI ozone retrievals are typically 0.3–0.5 in the lower troposphere over India.

## 2.2. GEOS-Chem simulations

We use the GEOS-Chem global CTM (v10-01; http://www.geos-chem.org) in this study. The model

includes a detailed mechanism of ozone-$NO_x$-VOC-aerosol tropospheric chemistry (Bey et al., 2001; Park et al., 2004; Mao et al., 2010, 2013) using the chemical kinetics recommended by Jet Propulsion Laboratory (JPL) and International Union of Pure and Applied Chemistry (IUPAC) (Sander, et al., 2011; IUPAC, 2013), and photolysis rates calculated by the Fast-JX scheme (Bian and Prather, 2002). Stratospheric ozone chemistry is represented by the linearized ozone parameterization (LINOZ)

(McLinden et al., 2000), and other stratospheric species are simulated using monthly averaged production and loss rates archived from the Global Modeling Initiative (GMI) model (Murray et al., 2013). Physical processes such as deposition and planetary boundary layer (PBL) mixing schemes are summarized in Table 1. The model has been applied in a number of studies on global and regional tropospheric ozone (Wang et al., 2013; Fiore et al., 2014; Zhang et al., 2014; Yan et al., 2016; Zhao et

al., 2017). A recent model evaluation with global tropospheric ozone datasets shows that GEOS-Chem





(v10-01) provides an improved ozone simulation relative to previous model versions (e.g., v8-01 in Zhang et al., (2010)) with no significant seasonal and latitudinal biases (Hu et al., 2017).

The model is driven by the Modern Era Retrospective-analysis for Research and Application (MERRA) assimilated meteorological fields (Rienecker et al., 2011). For input to GEOS-Chem, we downgrade the MERRA data to 2.5° longitude × 2° latitude and 47 vertical layers (extending from surface to 0.01 hPa) from the raw resolution of 0.667° longitude × 0.5° latitude and 72 layers. Emissions in the model are calculated using the Harvard-NASA Emission Component (HEMCO) (Keller et al., 2014). Year-specific anthropogenic emissions are from the Emissions Database for 150 Global Atmospheric Research (EDGAR v4.2 for emissions over 1990–2008; 2008 emissions are used for simulation afterwards) (EDGAR, 2011), overwritten with regional emission inventories as summarized in Table 1. Asian anthropogenic emissions are from the MIX emission inventory (Li et al., 2017).

Climate-sensitive natural ozone emissions such as biogenic non-methane volatile organic compounds (NMVOCs) emissions, lightning $NO_x$ emissions, soil $NO_x$ emissions are implemented in GEOS-Chem as summarized in Table 1. For the biomass burning emissions, we combine the inventory of the Atmospheric Chemistry and Climate Model Intercomparison Project (ACCIMP) (Lamarque et al., 2010) for 1990–1996 and the Global Fire Emission Database version 3 (GFED3) (van der Werf et al., 160 2010) for 1997–2010. Comparison of GFED3 and ACCMIP biomass burning CO emissions for their overlapping years (1997–2000) suggests ACCMIP is 30% higher. We thus reduce the 1990–1996 ACCMIP emissions by 30% to partly correct the gap between the two inventories. As atmospheric methane has a relatively long lifetime (about 9 years), its concentrations are prescribed in GEOS-Chem using year-specific measured concentrations from the NOAA Global Monitoring Division 165 (GMD) (see Table 1).




We conduct a standard simulation (BASE) with year-specific assimilated meteorology and anthropogenic emissions from 1990 to 2010 with the initial conditions generated by a two-year spin-up simulation. We also conduct sensitivity simulations by fixing one of the sources at the 1990 conditions, including anthropogenic emissions (FEMIS), global methane concentrations (FCH$_4$), and biomass burning emissions (FBIOB) as summarized in Table 2. Differences between the standard simulation and the sensitivity simulations are then used to estimate influences of interannual changes in the specific source on tropospheric ozone concentrations over India. All simulations are conducted for 1990–2010 as constrained by the availability of MERRA meteorology and emissions.

## 2.3. Ozone budgets diagnosed in GEOS-Chem

We analyze processes affecting lower tropospheric ozone budgets in each model grid including ozone chemical production and loss, horizontal and vertical transport, and dry deposition. These processes are diagnosed at every hour and averaged to monthly mean. Net productions are calculated as the differences between ozone chemical production and loss rates. Horizontal transport for each grid is the net horizontal flux from or to adjacent grids. Vertical transport is estimated as the flux at the top of the lower troposphere (600 hPa in this study) with positive values representing downward transport.

## 3. Seasonal variation of lower tropospheric ozone over India

## 3.1. Variations of meteorology and emissions

Variations in tropospheric ozone are subject to changes in precursor emissions and meteorology conditions such as local temperature and transport pattern. We show in Figure 1 and 2 spatial and seasonal variations in MERRA meteorological variables (surface temperature, 850 hPa specific humidity (SPHU), and cloud cover), as well as anthropogenic NO emissions, and biomass burning CO emissions over India averaged for the 5-year period (2006-2010). Meteorological conditions in India



have distinct seasonal variations associated with the monsoon onset and retreat. Temperature increases from winter (January) to late spring (May) with increasing solar radiation. The onset of summer monsoon in late May brings moist air from oceans and drives strong air convergence and uplift over India, which lead to cloudy conditions, large decreases in surface temperature (about 8 °C from May to

August), and enhancements in SPHU (5 g kg$^{-1}$) (Figure 1 and 2a). Surface temperature and SPHU become relatively stable with the retreat of summer monsoon since September, and then both decrease in winter when the winter monsoon brings cold and dry air.

Figure 1 and 2b also show anthropogenic NO emissions of 4.51 Tg a$^{-1}$ (per annum) in India, with

emissions in winter (December, January, and February) 4.2% higher than summer (June, July, and August) due to more active residential heating. Anthropogenic emissions are higher over the northern India including the Indo-Gangetic Plain (IGP, extending from the plain of the Indus River to the plains of the Ganges River) and southern India, following the distribution of population density (Beig and Brasseur, 2006; Kumar et al., 2012). Biomass burning emissions in Southeast Asia are active in March

and April (Figure 1e) that account for 62% of the annual biomass burning CO emissions in India (Figure 2b), and are likely due to open burnings during post-harvesting seasons as agricultural field clearance (Venkataraman et al., 2006). Hot and dry air conditions in March and April also likely enhance the wildfire frequency and strength (Westering et al., 2006; Jaffe et al., 2008; Lu et al., 2016).

**3.2. Variations in the pre-summer monsoon season**

Figure 2c shows OMI observed and GEOS-Chem model simulated seasonal variations of lower tropospheric ozone concentrations averaged over India and over the 5-year (2006–2010) period. Figure 3 shows their spatial distributions. Model results are applied with OMI averaging kernel matrices and a priori profiles. OMI shows annual mean lower tropospheric ozone concentration of 45.9 ppbv over

India with maximum (54.1 ppbv) in the pre-summer monsoon season and minimum (40.5 ppbv) in

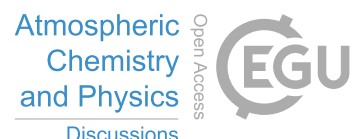

July–August when the summer monsoon reaches its annual strongest stage. A similar seasonal cycle of lower tropospheric ozone has been found in the Tropospheric Emission Spectrometer (TES) observations (Kumar et al., 2012). Spatially, observed ozone concentrations are higher in northern India and IGP regions than southern regions, consistent with reported surface measurements (Lal et al.,

2000; Naja and Lal, 2002; Beig et al., 2007; Reddy et al., 2008; Ojha et al., 2012; Kumar et al., 2012). While model results are about 2 ppbv (4.4%) higher annually with the largest overestimate occurring in January–May, they generally capture the seasonal and spatial variations of OMI observed lower tropospheric ozone concentrations ($r$ = 0.81–0.97). Comparison of GEOS-Chem surface ozone concentrations for the year 2010 with measurements at 6 Indian surface sites reported by Sharma et al.

(2016) also shows consistent seasonal variations but with positive biases of about 8 ppbv (Supplemental Figure S1; 43.5 ± 7.4 ppbv in the model vs. 35.0 ± 8.4 ppbv in measurements). Similar model overestimates over India are reported in Kumar et al. (2012) using the WRF-Chem and MOZART models, and are likely due to uncertainties in $NO_x$ emissions and the coarse model resolution.


Figure 2c and Figure 3 also identify the processes affecting lower tropospheric ozone burden over India. Here we separate our analysis below to four time periods: pre-summer monsoon seasons (March–April), summer monsoon seasons (May–August), post-summer monsoon seasons (August–October), and wintertime (November–following March). In March and April, the mean lower

tropospheric ozone concentration over India increases by 9.8 ppbv from the wintertime that can be explained by significant enhancements of ozone production (from 5.1 Tg in January to 10.9 Tg in April) and net production (from 2.9 to 5.0 Tg month$^{-1}$). As anthropogenic emissions slightly decrease from wintertime, ozone production enhancements are more likely associated with ozone-favorable weather conditions such as stronger solar radiation and increasing temperature. These changes not only

enhance ozone photochemistry efficiencies in the presence of $NO_x$ (Jacob and Winner, 2009; Doherty

et al., 2013; Psuede et al., 2015), but also increase natural emissions such as biogenic VOCs and soil $NO_x$ emissions. We find that biogenic isoprene and soil $NO_x$ emissions over India as calculated in the model are about a factor of 2 higher in the pre-summer monsoon season than wintertime. Additional ozone enhancements are due to intense biomass burning emissions. We can see strong ozone

production in central eastern India in the pre-summer monsoon season (Figure 3c) associated with biomass burning regions (Figure 1e).

Horizontal transport (both west-east and north-south transport) decreases Indian lower tropospheric ozone in March–April (Figure 2c). As shown in Figure 3a, an anti-cyclonic wind pattern dominates the

Indian subcontinent during January–April. Prevailing northeastern winds in northern India efficiently transport the ozone rich-air downwind, and circulate into the southern India, resulting in a deficient budget in northern India but positive in southern India (Figure 3d and Supplemental Figure S2). This low tropospheric transport pattern implies that southern India would likely suffer ozone pollution transported from the ozone-rich northern India in winter and pre-monsoon seasons. Vertical transport

at 600 hPa has a relatively small positive contribution (1.8 Tg month$^{-1}$) to the lower tropospheric ozone budget over India in March–April, partly due to offset of downward import over northern India and upward export over southern India (Figure 3e). Northern India with higher elevation is likely subject to more stratospheric ozone influences in the period as evidenced by ozonesonde observations (Kumar et al., 2010; Ojha et al., 2014), while southern India is usually characterized by strong air uplift through

convection.

### 3.3. Variations in the summer monsoon season

The monthly mean lower tropospheric ozone concentration over India decreases from 54 ppbv in May to the seasonal minimum of 40.5 ppbv in August. We find that ozone decrease starts earlier in southern

India than in northern India, as can be seen from Figure 3a that ozone concentrations in the south are

higher in March–April than May–June, while ozone in the north reaches its annual maximum in May–June. These patterns can be explained by the temporal differences in arrivals of summer monsoon (Kumar et al., 2012).

The onset of South Asian summer monsoon in late May is usually diagnosed with the prevailing westerly winds from Arabian Seas to Bay of Bengal (Wang and Lin, 2002; Gadgil, 2003), leading to cloudy and rainy weather conditions over the Indian subcontinent. Rainfalls efficiently remove ozone precursors as shown from satellite observation of $NO_2$ column (Kumar et al., 2012). Weak solar radiation and low temperature are not favorable for ozone photochemical formations. In addition,

water vapor from marine air serves as chemical loss of ozone at low $NO_x$ conditions (Jacob et al., 2000). The onset of summer monsoon also brings strong air convergence and uplift as indicated by the large-scale air upward velocity in May–August (Figure 3e). Biomass burning emissions are negligible in the summer monsoon season, and anthropogenic emissions also reach its annual minimum. These changes together lead to declines in monthly ozone chemical production by 4.2 Tg (from 11.8 Tg in

May to 7.6 Tg in August) as integrated over India.

Comparable to chemical production, changes in vertical transport also show a large contribution on the decline of lower tropospheric ozone over India in the summer monsoon season. The monthly vertical transport flux at 600 hPa integrated over India is near zero in May, and reaches −3.3 Tg in August,

offsetting the net ozone production in this month (2.9 Tg). Strong vertical convection in July–August effectively uplifts ozone pollution from the lower troposphere to the upper troposphere that can then be carried by the easterly jet to other parts of the world such as Mediterranean (Park et al., 2007; Lawrence and Lelieveld, 2010; Lal et al., 2013), affecting the global tropospheric ozone distribution. We find that horizontal transport from the ocean can lower ozone over northwestern India, especially

in May and June when the summer monsoon onsets, consistent to previous observations (Srivastava et





al., 2012b; Lal et al., 2014). Horizontal transport also enhances lower tropospheric ozone concentrations in eastern India and the Bay of Bengal (Figure 3d). The overall contribution of horizontal transport to the Indian lower tropospheric ozone budget is thus relatively small in July–August (0.91 Tg month$^{-1}$) relative to vertical export (Figure 2c).


### 3.4. Variations in the post-monsoon season and wintertime

Lower tropospheric ozone concentrations over India slightly increase since September and reach a second peak in October, associated with increases in precursor emissions and decreases in upward transport. With the southward movement of solar radiation and the summer monsoon retreatment, both surface temperature and lower tropospheric SHPU show decreasing patterns (Figure 1), leading to reductions in ozone chemical loss. In addition, retreatment of the summer monsoon reduces air uplift over the Indian subcontinent (Figure 3e), which allows 2.8 Tg more ozone maintaining in the Indian lower troposphere in October compared to September (Figure 2c).



In November to the following March, ozone production in the Indian lower troposphere reaches its annual minimum (4.7 Tg month$^{-1}$ in December) due to low temperature conditions. Horizontal wind patterns are similar to those in the pre-summer monsoon season with a large negative contribution (–6.1 Tg month$^{-1}$ in November–March) to the Indian lower tropospheric ozone budget. The emergence of winter monsoon leads to large-scale air subsidence over northern India, with a total import flux of 1.5 Tg month$^{-1}$ in winter. The low ozone net production and strong horizontal export result in relatively low ozone levels in wintertime (43.6 ppbv in the lower troposphere). It should be noted that some observational studies reported the highest surface ozone concentrations at several urban or semi-urban sites in southern India (e.g., Ahmedabad, 23°N, 73E; Pune, 18°N, 74°E; Trivandrum, 8°N, 77°E) in wintertime instead of the pre-summer monsoon season likely due to high local precursor emissions (Beig et al., 2007; David and Nadir, 2011; Kumar et al., 2012; Lal et al., 2014; Sahu et al., 2014).







Ozonesonde observations at the Ahmedabad and Hyderabad airports (Lal et al., 2014; Sahu et al., 2014) indicate that ozone concentrations in the free troposphere (near 5 km) are higher in the pre-summer monsoon season than in winter, consistent with OMI observations in our study.

**4. Interannual variability of lower tropospheric ozone over India**

**4.1. Correlation with surface temperature in pre-summer monsoon seasons**

We now analyze interannual variability of lower tropospheric ozone in India with focus on pre-summer monsoon seasons when concentrations are highest and summer monsoon seasons when concentrations are subject to monsoon variability. As have been discussed in Sect. 3.2, tropospheric

ozone concentrations in the pre-summer monsoon season are largely controlled by ozone production enhancements due to ozone-favorable weather conditions such as high temperature, and are likely amplified by biomass burning emissions. We also find here strong interannual correlations between surface temperature and lower tropospheric ozone concentrations over India. Figure 4 shows that 9–year (2006–2014) time-series of OMI observations have a positive correlation ($r = 0.55$) with MERRA

surface temperature in pre-summer monsoon seasons. GEOS-Chem model results (the BASE simulation) for the period of 1990–2010 capture this positive correlation ($r = 0.58$), and the correlation persists when both variables are detrended ($r = 0.50$).

Figure 5 shows the spatial distribution of correlation coefficients between the lower tropospheric

ozone concentration (OMI vs. GEOS-Chem) and surface temperature. Stronger correlations ($r \approx 0.8$) are found in northern India (e.g., the IGP regions) and southern India where $NO_x$ emissions are high. The dependence of the ozone–temperature relationship on $NO_x$ emission levels is consistent with previous studies reflecting higher ozone formation potential over high $NO_x$ regions (Jacob and Winner, 2009; Doherty et al., 2013; Psuede et al., 2015). As pointed out by Kumar et al. (2012) and Sharma et

al. (2016), ozone production in most regions of India is $NO_x$ limited. Reduction of $NO_x$ emissions over



those regions during pre-summer monsoon seasons can not only lower ozone chemical production, but also weaken the sensitivity of ozone to potential future warming.

We examine the overall sensitivity of the ozone–temperature correlation in India to emissions. As
shown in Figure 4, the sensitivity simulation with fixed anthropogenic emissions (FEMIS) shows a slightly lower correlation ($r = 0.54$). It indicates that despite the dependence of ozone–temperature correlations on $NO_x$ emission levels regionally as shown above, the mean correlation over India is still dominated by the temperature impact on ozone chemical production. We also calculate the interannual variability contributed by biomass burning emissions as ozone differences between the BASE
simulation and the FBIOB simulation. Biomass burning emissions have a large interannual variability with CO emissions ranging from 0.97 Tg to 4.7 Tg over 1990–2010 (Supplemental Figure S3). As can be seen from Figure 4, the ozone interannual variability contributed by biomass burning emissions is weakly correlated with the BASE lower tropospheric ozone ($r = 0.29$). However, they are important in high ozone and high temperature years. In years such as 1999 and 2010, biomass burning caused 1.5–
2.2 ppbv higher ozone, enhancing the variability of lower tropospheric ozone. This feature has also been found in the western US (Jaffe et al., 2008; Lu et al., 2016) as high temperature conditions favor both biomass burning emissions and ozone production, and thus amplify lower tropospheric ozone concentrations.

## 4.2. Impact of monsoon strength in summer monsoon seasons
We have also shown above that lower tropospheric ozone concentrations over India vary associated with the onset and retreat of the South Asian summer monsoon. The interannual variability of lower tropospheric ozone over India in the summer monsoon seasons (May–August) can then be affected by the strength of the South Asian summer monsoon. To quantify this relationship, we calculate the
monsoon strength using a monsoon index that is proposed by Li and Zeng (2002) and has been applied

to quantify impacts of the East Asian monsoon on air pollution over China (Zhu et al., 2012; Yang et al., 2014). The monsoon index ($\delta$) for grid $(i, j)$ in the Northern Hemisphere in the month $m$ and year $y$ is given as:

$$\delta_{y,m}(i,j) = \frac{\|\overline{V}_1 - \overline{V}_{y, m}\ (i,j)\|}{\|(\overline{V}_1 + \overline{V}_7)/2\|} - 2 \qquad (1)$$

where $\overline{V}$ represents monthly mean wind speed from the MERRA dataset, and $\overline{V}_1$ and $\overline{V}_7$ are climatological (1990-2010 in our study) monthly wind speed in January and July, respectively. The norm of a given variable A is defined as:

$$\|A\| = \left(\iint |A|^2 dS\ \right)^{1/2} \qquad (2)$$

where $S$ represents the spatial domain for integration. Details for the calculation of $\|A\|$ are given in
Li and Zeng (2002). In this study, we use the region of 35°E–90°E, 5°N–35°N at 850 hPa and over May–August for the South Asian summer monsoon index (SASMI).

Figure 6 shows the time series of SASMI anomalies relative to the 1990–2014 climatology, and their correlations with OMI observed and model simulated lower tropospheric ozone concentration
anomalies over India for the summer monsoon seasons. Positive and negative SASMI values represent strong and weak summer monsoons, respectively. We find no significant trend in the South Asian summer monsoon strength over 1990–2014. Significant negative correlations between Indian lower tropospheric ozone concentrations and SASMI values can be seen for both OMI observations ($r = -$0.46, 2006–2014) and GEOS-Chem BASE results ($r = -0.52$, 1990–2010). Removing interannual
changes in anthropogenic emissions (FEMIS) results in a stronger correlation of –0.72, reflecting the dominant role of monsoon strength on the interannual variability of Indian lower tropospheric ozone in summer monsoon seasons. We find that the correlations between SASMI and simulated ozone are even stronger at the surface level ($r = -0.72$ for BASE and $r = -0.83$ for FEMIS; figure not shown).



Yang et al. (2014) previously found positive correlations between the East Asian summer monsoon

strengths and surface ozone concentrations over mainland China. They attributed higher surface ozone

in stronger summer monsoon years to a smaller outflow of ozone to the East China Sea. Our results

show an opposite response of lower tropospheric ozone to summer monsoon strengths over India. To

understand the negative correlations, we illustrate in Figure 7 the differences in meteorological

variables, lower tropospheric ozone concentrations, and relevant processes between weak and strong

monsoon conditions (represented by averages over 5 years with the lowest and the highest SASMI

over 1990–2010, respectively). We focus on model results from the FEMIS simulation to exclude the

influence from interannual changes in anthropogenic emissions. We find that lower tropospheric ozone

concentrations averaged over India are 3.4 ppbv higher in weak summer monsoon years than those in

strong monsoon years. Weak summer monsoon conditions show higher surface temperature (1.1 °C),

drier air (–0.5 g kg$^{-1}$), and lower cloud cover over India, together accounting for a higher ozone net

production of 0.4 Tg month$^{-1}$ than the strong summer monsoon conditions (Figure 7). In addition,

weaker convergence and convection in weak summer monsoon years can cause the total upward ozone

flux 0.2 Tg month$^{-1}$ smaller, but this is closely offset by stronger horizontal outflows. Together, we

find that differences in ozone net production are the key factor explaining differences in the lower

tropospheric ozone burden over India between the strong and weak summer monsoon years.

## 5. Long-term trend and contributing drivers

As for the long-term trend in the Indian lower tropospheric ozone, the 9–year OMI observations

appear to be too short to provide a long-term trend estimate. As can be seen from Figure 4 and Figure

6, OMI observed mean Indian lower tropospheric ozone concentrations over 2006–2014 show large

positive trends of $0.42 \pm 0.38$ ppbv year$^{-1}$ (mean $\pm$ 95% confidence level, p-value = 0.03) for the pre-

summer monsoon seasons and $0.58 \pm 0.71$ ppbv year$^{-1}$ (p-vale = 0.09) for the summer monsoon

seasons. However, these 9-year trends are mainly driven by the low values in years 2006–2008. It



should be acknowledged that the OMI dataset are likely influenced by the OMI row anomaly that potentially results in overestimates in ozone trends over the tropics (Huang et al., 2017a).

Here we analyze the long-term trends over 1990–2010 simulated by the GEOS-Chem model. Figure 8 shows the spatial distribution of simulated lower tropospheric ozone trends for the annual average, as

well as for averages in pre-summer monsoon seasons and summer monsoon seasons. Annually, the Indian lower tropospheric ozone is increasing at a statistically significant rate of $0.19 \pm 0.07$ (p-value < 0.01) ppbv year$^{-1}$. Larger ozone trends ($0.27 \pm 0.12$ ppbv year$^{-1}$, p-value < 0.01) are shown in pre-summer monsoon seasons than those in summer monsoon seasons ($0.16 \pm 0.14$ ppbv year$^{-1}$).

The sensitivity simulations allow us to quantify potential ozone trend drivers, including changes in anthropogenic emissions, biomass burning emissions (Supplemental Figure S3), and global methane concentrations. Figure 8 also shows contributions from each factor calculated as differences in trends between the BASE simulation and the sensitivity simulations. Changes in anthropogenic emissions largely explain the increasing trends in the lower tropospheric ozone over India, which account for

0.18, 0.21, and 0.19 ppbv year$^{-1}$ for the annual, pre-summer monsoon seasonal, and summer monsoon seasonal means, respectively. Global methane concentration increases also contribute small increases of 0.02 ppbv year$^{-1}$ in the lower tropospheric ozone, and the contributions are larger in the middle and upper troposphere (figure not shown). Biomass burning emissions in East and South Asia show a decreasing trend over 1990–2010 (Figure S3) that results in small negative trend contributions in the

lower tropospheric ozone. As anthropogenic emissions in India are projected to rise in the future (Ghude et al., 2016), we may expect further increases in the Indian tropospheric ozone. Continuous ozone monitoring measurements are required to better quantify long-term changes in tropospheric ozone over India.



## 6. Conclusions

In summary, we have investigated the processes controlling seasonal and interannual variations of lower tropospheric ozone concentrations over India and their linkages to the South Asian summer monsoon. We use OMI satellite observations of lower tropospheric ozone over 2006–2014 and GEOS-Chem global model simulations over 1990–2010 driven by assimilated meteorological fields and best-known emissions to better quantify the controlling processes.

Both OMI satellite observations and GEOS-Chem simulations show that ozone in the Indian lower troposphere (surface to 600 hPa) peaks in the pre-summer monsoon season (March–April, 54.1 ppbv), and decreases dramatically to the annual minimum (40.5 ppbv) during the summer monsoon season (May–August). It then re-rises in the post-summer monsoon season (September–October), and flattens in winter. GEOS-Chem process analyses on the Indian lower tropospheric ozone budget indicate that the pre-summer monsoon seasonal ozone maximum is mainly driven by enhanced ozone chemical production due to favorable meteorological conditions (strong solar radiation with low cloud cover, high temperature, and relatively dry air), as well as active biomass burning emissions in spring. We find that overall horizontal transport is important for ventilating lower tropospheric ozone, while vertical transport has a small positive contribution on the lower tropospheric ozone budget over India in the pre-summer monsoon season.

The onset and evolution of the summer monsoon in May–August brings low temperature, weak solar radiation conditions and moist air from the Arabian Seas, leading to significant reduction of ozone production (–4.2 Tg month$^{-1}$ from May to August) over India. We also highlight the contribution of upward transport on the Indian lower tropospheric ozone budget in July–August (–3.3 Tg month$^{-1}$), which is comparable to the change in ozone production, and potentially transport Indian ozone to other parts of the world. In the post-summer monsoon season (September–November), lower tropospheric



ozone over India re-rises due to weakening ozone upward transport associated with the summer monsoon retreat. In winter, low temperature conditions limit the ozone production, and strong horizontal outflows largely lower the ozone burden over India.

We show that interannual variability of lower tropospheric ozone over India in pre-summer monsoon
and summer monsoon seasons are strongly linked to climate variability. Both OMI observed and model simulated lower tropospheric ozone in pre-summer monsoon seasons are significantly correlated with surface temperature ($r = 0.55$–$0.58$). Higher ozone–temperature correlations ($r > 0.7$) are found over high $NO_x$ emission regions. Lower tropospheric ozone in summer monsoon seasons are strongly influenced by the South Asian monsoon strengths. Comparing the 5 weakest South Asian
summer monsoon years with the 5 strongest monsoon years, we find that lower tropospheric ozone over India are 3.4 ppbv higher in the weakest monsoon years, mainly because of higher temperature, drier air, and lower cloud cover that enhance ozone production as well as less ozone vertical export. These interannual variations indicate that lower tropospheric ozone concentrations in India are potentially affected by decadal climate variability such as the El Niño - Southern Oscillation (Kumar et
al., 1999) and AMO (Lu et al., 2006).

We have also analyzed the long-term trends in lower tropospheric ozone over India and their drivers as suggested by the GEOS-Chem model. Model results over 1990–2010 show an annual mean increasing trend of $0.19 \pm 0.07$ ppbv year$^{-1}$ over India, which is mainly driven by rising anthropogenic emissions
with small contributions ($0.02$ ppbv year$^{-1}$) from global methane concentration increases. Our study emphasizes the importance to understand tropospheric ozone changes and drivers at multiple time scales in India. Ozone pollution in India may become more severe with rises in anthropogenic emissions and population, and potentially exert large impacts on the global tropospheric ozone distribution due to frequent deep convection over South Asia. Analyses of long-term ozone



measurements in India are in need to better understand their variations and associated environmental effects.

## Data availability

The datasets including measurements and model runs can be accessed from websites in the reference

list or by contacting the corresponding author (Lin Zhang; zhanglg@pku.edu.cn).

## Acknowledgements

This work is supported by the National Natural Science Foundation of China (41475112) and China's

National Basic Research Program (2014CB441303). Xiao Lu is also supported by the Chinese

Scholarship Council. The authors thank Prof. Daniel Jacob at Harvard University for the useful

comment. The authors acknowledge the Harvard GEOS-Chem Support Team for the model

maintenance and development.

**3 Figures are included in the supplement related to this article.**

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



**Table 1.** A summary of physical processes, anthropogenic and natural emissions used in GEOS-Chem

| | Descriptions | Sources and references |
|---|---|---|
| **Physical processes** | | |
| Wet deposition | Parameterization for scavenging in both convection and large-scale precipitation for soluble gases and aerosols | Mari et al. (2000); Liu et al. (2001); Amos et al. (2012); |
| Dry deposition | Resistance-in-series algorithm | Wesely (1989); Zhang et al. (2001) |
| PBL mixing | Non-local mixing scheme | Lin and McElroy (2010) |
| **Anthropogenic emissions** | | |
| Global | Emissions Database for Global Atmospheric Research (EDGAR v4.2) | EDGAR (2011) |
| East Asia and South Asia | MIX emission inventory | Li et al. (2017) |
| United States | Environmental Protection Agency (EPA) National Emission Inventory (NEI) | US EPA-NEI (2015) |
| Canada | Canadian Criteria Air Contaminant inventory | |
| Europe | European Monitoring and Evaluation Program (EMEP) | Vestreng and Klein (2002) |
| Mexico | Big Bend Regional Aerosol and Visibility Observational study inventory (BRAVO) | Kuhns et al. (2005) |
| **Natural sources** | | |
| Biogenic emissions | Model of Emissions of Gases and Aerosols from Nature (MEGAN) | Guenther et al. (2006) |
| Lightning ($NO_x$) emissions | Parameterization based on cloud top height, and spatially constrained by satellite observed lightning flashes | Price and Rind (1992); Sauvage et al. (2007); Murray et al. (2012) |
| Soil $NO_x$ emissions | empirical parameterization of available nitrogen (N) | Hudman et al. (2012) |
| Biomass burning emissions | Atmospheric Chemistry and Climate Model Intercomparison Project (ACCIMP) for 1990–1996 and Global Fire Emission Database version 3 (GFED3) for 1997–2010 | Lamarque et al. (2010); van der Werf et al. (2010) |
| Methane | Prescribed over four latitudinal bands with year-specific mixing ratios constrained by | |

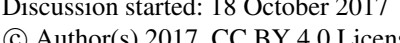



measurements from the NOAA Global Monitoring Division (GMD). Concentration ranges over 1990−2010 are given below:

90°−30°S (1663−1732 ppbv), 30°S−0° (1666−1741 ppbv), 0°−30°N (1733−1801 ppbv), and 30°−90°N (1792−1855 ppbv)


**Table 2.** Configuration of the GEOS-Chem simulations. 'V' indicates that specific inputs vary interannually in the simulation, and '1990' denotes that the inputs are fixed to 1990 conditions.

| Simulation | BASE | FEMIS | FBIOB | FCH$_4$ |
|---|---|---|---|---|
| Anthropogenic emissions | V | 1990 | V | V |
| Biomass burning emissions | V | V | 1990 | V |
| Global methane concentrations | V | V | V | 1990 |




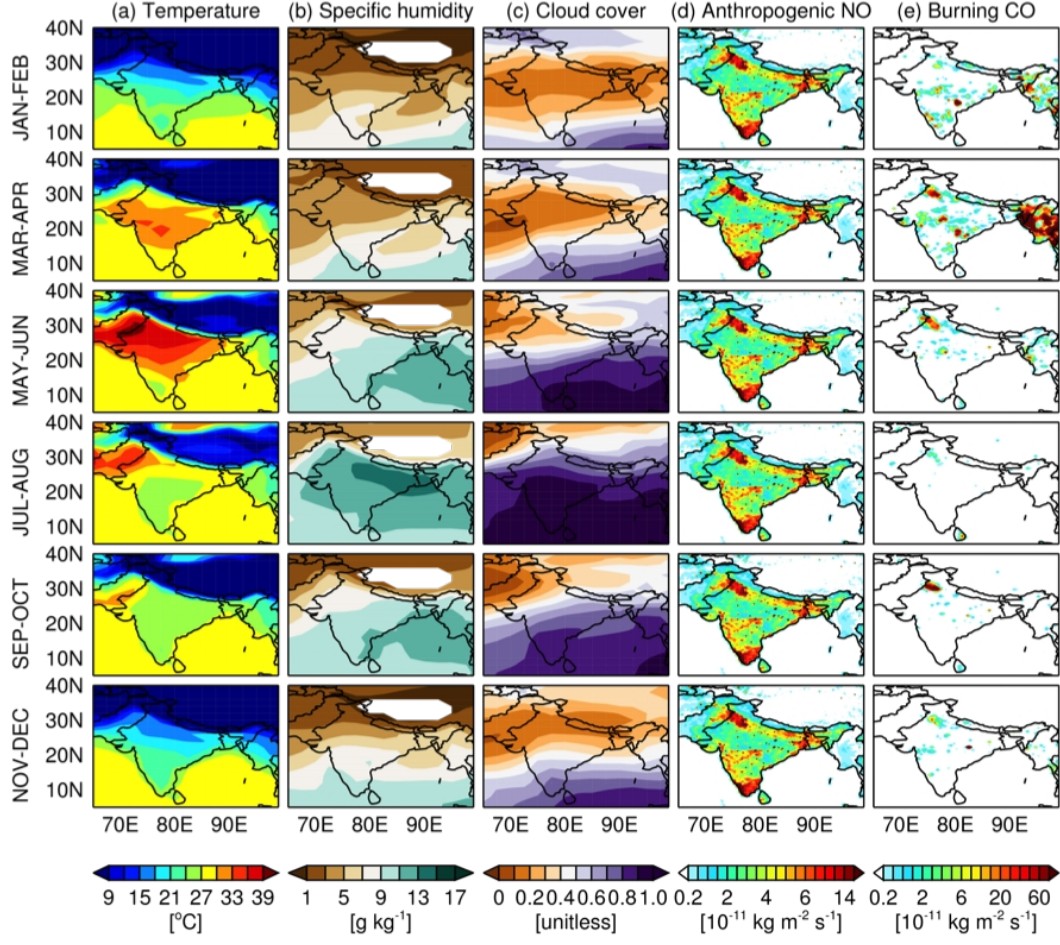

**Figure 1.** Spatial distributions of bimonthly mean (a) surface temperature, (b) 850 hPa specific
humidity, (c) cloud cover, (d) anthropogenic NO emissions, and (e) biomass burning CO emissions
averaged for 2006–2010.




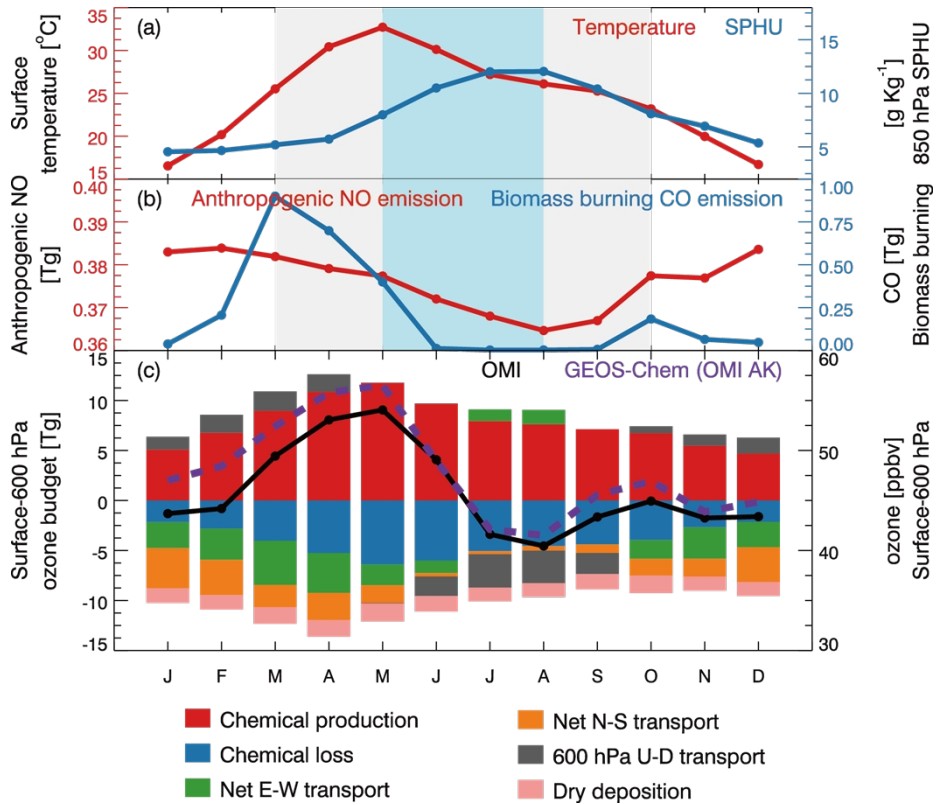

**Figure 2.** Monthly mean (averaged for 2006–2010) (a) surface temperature (red) and 850 hPa averaged specific humidity (SPHU, blue), (b) anthropogenic NO emissions (red) and biomass burning CO emissions (blue), and (c) lower tropospheric ozone (averaged for surface to 600 hPa, in unit of ppbv) from OMI satellite observations (black) and GEOS-Chem model simulations with OMI averaging kernel matrices and a priori profiles applied (dashed purple). The shading in panel (a) and (b) represents pre, during, and post summer South Asian monsoon periods. Color bars in panel (c) show processes that affect lower tropospheric ozone budget over the Indian domain diagnosed in GEOS-Chem simulations.





**Figure 3.** Spatial distributions of bimonthly mean lower tropospheric ozone from (a) OMI satellite observations and (b) GEOS-Chem simulations (with OMI averaging kernel matrices and a priori profiles applied). Also shown are changes in lower tropospheric ozone burden contributed by (c) net chemical production, (d) net horizontal transport, and (e) vertical transport flux at 600 hPa. All values are averaged for 2006–2010. Wind patterns are overlaid in (a). White dots in (e) denote model grid cells where mean absolute vertical velocities at 600 hPa exceed 5 mm s$^{-1}$.





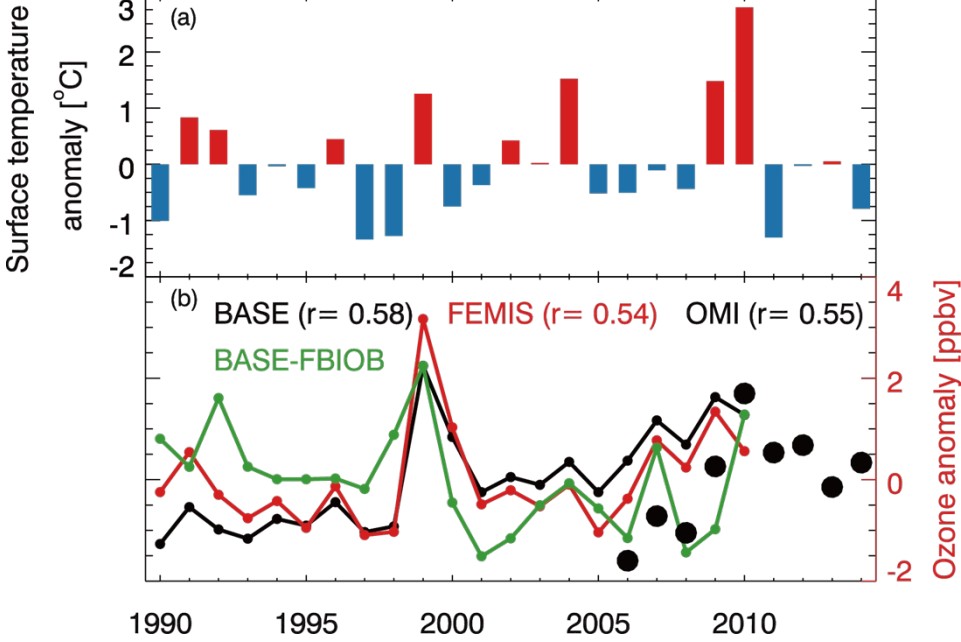

**Figure 4.** Time series of mean (a) surface temperature and (b) lower tropospheric ozone anomaly averaged for pre-summer monsoon seasons. OMI observed ozone anomalies are shown in black circles, and model results from the BASE and FEMIS simulations are shown in black and red lines, respectively. The green line shows the ozone anomaly contributed by biomass burning emissions. Correlation coefficients (r) between surface temperature and ozone are shown inset.





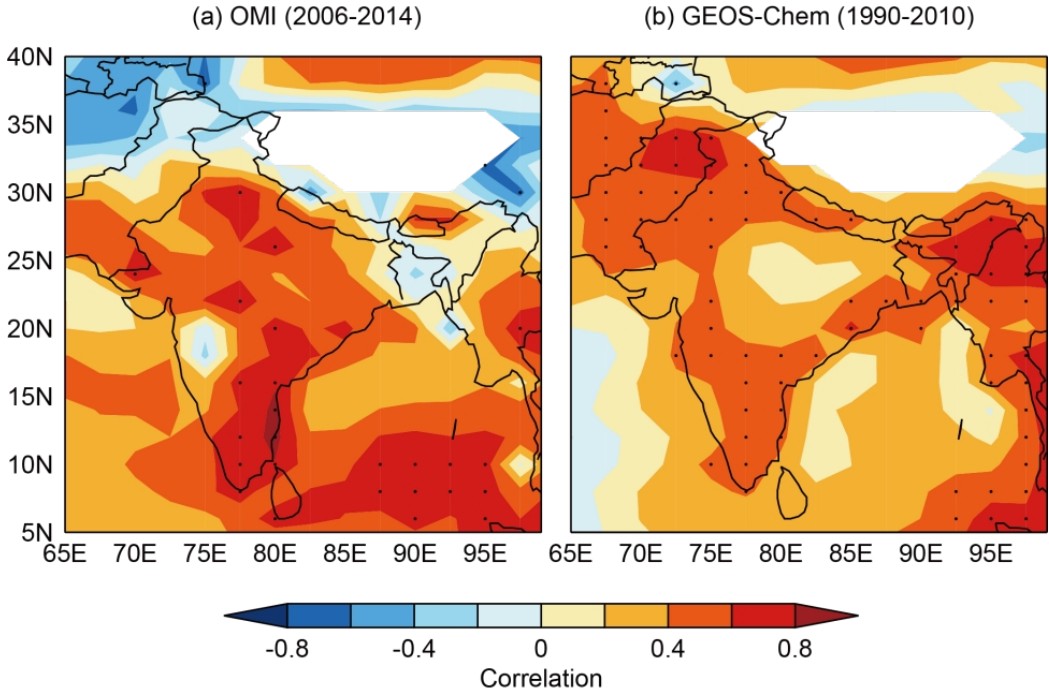

**Figure 5.** Spatial distribution of correlation coefficients between surface temperature and lower tropospheric ozone in pre-summer monsoon seasons from (a) OMI satellite observations for 2006–2014 and (b) the GEOS-Chem BASE simulation for 1990–2010. Black dots denote statistically significant (p-value < 0.05).



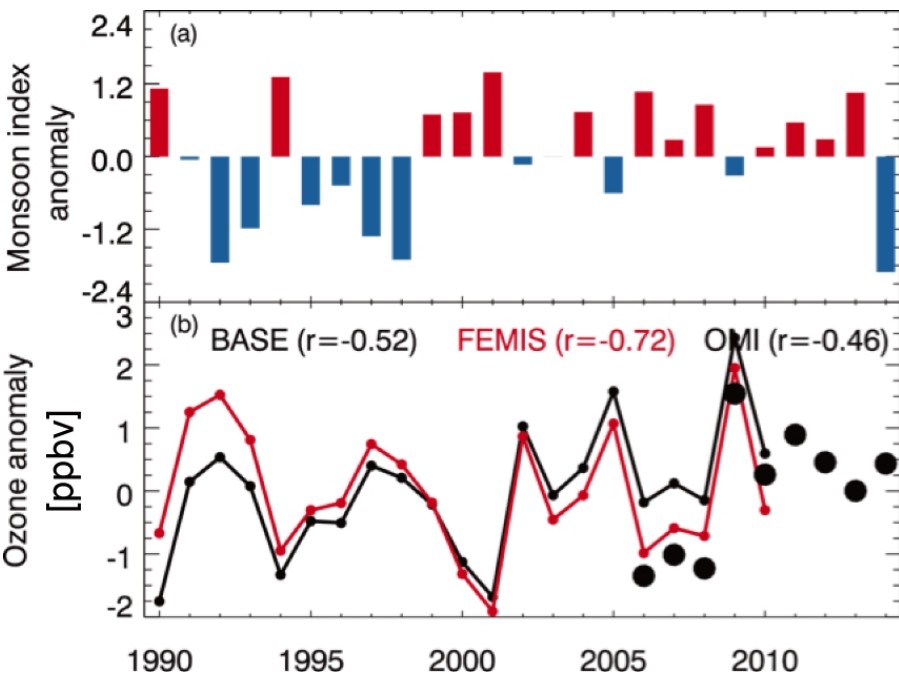

**Figure 6.** Time series of (a) South Asian summer monsoon index and (b) lower tropospheric ozone
averaged over India in summer monsoon seasons. Values are anomalies over 1990–2010. OMI
observed ozone anomalies are shown in black circles, and model results from the BASE and the
FEMIS simulation are shown in black and red lines, respectively. Correlation coefficients with the
monsoon index are shown inset.

925



**Figure 7.** Differences in May–August monthly mean (a) lower tropospheric ozone concentration, (b) surface temperature, (c) 850 hPa specific humidity, (d) lower tropospheric net ozone production, (e) lower tropospheric net horizontal transport with wind vectors overplotted, and (f) vertical transport at 930 600 hPa between the weak and strong summer monsoon years (weak minus strong). Values inset are averages (a–c) or totals (d–f) over the Indian land. Red dots in (f) denote regions with stronger air uplift in weak compared to strong summer monsoon years.





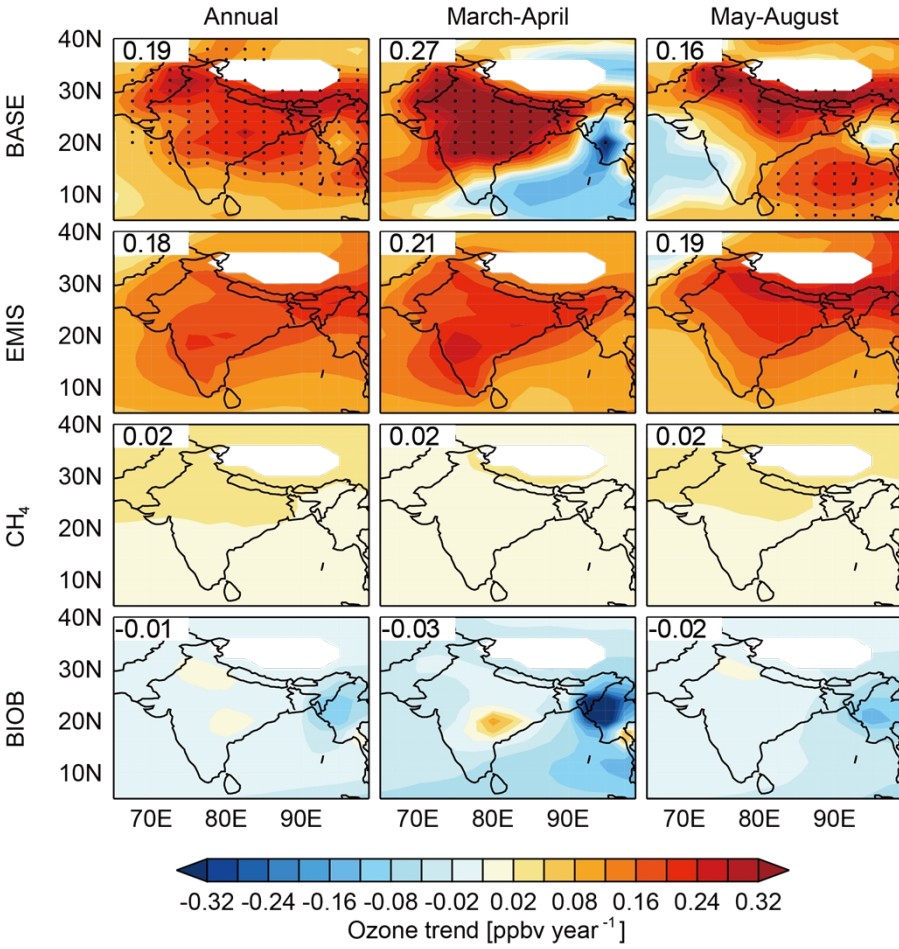

**Figure 8.** Simulated (1990–2010) trends in lower tropospheric ozone and factors contributing to simulated trends. Trends are calculated for annual averages (left column), pre-summer monsoon seasons (central column), and summer monsoon seasons (right column). Values inset are mean trend in ppbv year$^{-1}$ averaged over the Indian land. Black dots in the first rows denote significant trends in BASE simulations. Trends contributed by interannual changes in anthropogenic emissions (EMIS), global methane concentrations (CH$_4$), and biomass burning emissions (BIOB) are estimated.