# Peer review of "Lower tropospheric ozone over India and its linkage to the South Asian monsoon"

_Atmospheric Chemistry and Physics, 2017_

## Referee Comment (RC1) · Anonymous Referee #1 · 29 Nov 2017

The manuscript by Lu et al. investigated the lower tropospheric ozone over India and its linkage to the south Asian Monsoon by using satellite observations and GEOS-Chem model. Spatial and temporal characteristics of lower tropospheric ozone over India were analyzed in terms of seasonal cycle and inter-annual variability (for 2006-2010) and also long term-trends (for 1990-2010). The contribution/roles of different processes, including precursor emissions (anthropogenic NOx, biomass burning etc), meteorology (horizontal and vertical transport), chemical production, and dry deposition were discussed. The linkage of lower tropospheric ozone concentrations over India to the south Asian Monsoon were also quantified.

This comprehensive analysis focus on the ozone over Indian region where the precursor emissions are still rising in recent years. This study is thorough and clear, which would help to understand the local and global environment effects of India ozone. Overall, this manuscript is well organized, states the problem, outlines the model experiments, and describes the model results. This study fits the scope of ACP. I recommend publication.

Below are several comments that I think the authors may address to improve the manuscript.

General comments:
1. Line 162-163. The authors reduced the 1990-1996 ACCMIP emissions by 30% to correct the gap between GFED3 and ACCMIP, and also to get a full set of data for 1990-2010. It's quite understandable and straightforward. While the readers might get the impression that GFED3 is more accurate than ACCMIP. I'm just curious is there any reference that evaluate the two biomass burning emission inventories for this region? Any discussions of the sensitivity of the biomass burning emissions would be helpful.

2. Line 177. The contributions of different processes are analyzed, including chemical production and loss, horizontal and vertical transport and dry deposition. I'm wondering if the cloud-chemistry/wet scavenging are considered as the two processes might involve the O3 chemistry. Or are they already considered but just ignored as the contribution are too small? Some discussion would be helpful.

3. Line 181. For the horizontal transport (E-W, N-S) in the text and also figure 2, it would be better to give the directions for positive/negative values.

4. Section 3.1. The anthropogenic NOx emissions and biomass burning emissions are shown in this section while no discussions of NMVOC emissions are provided. While anthropogenic and biogenic NMVOC are also important ozone precursors. Especially for biogenic NMVOC, it might have strong seasonal cycle which might change the NOx/VOC regime in different seasons. Some discussions would he necessary here.

5. Line 213. When retrieving the OMI observed tropospheric ozone, some discussions of the sensitivity of the priori profile and average kernel matrices would be helpful. Are there any system biases or different biases in different seasons? Any uncertainties?

6. Line 242. It's quite important to state that biogenic isoprene and soil Nox emissions are higher in the pre-summer monsoon than wintertime which help the reader to understand the results. As suggested in Q4, it will be great to give a table that lists the seasonal/annual emissions from different sources (biogenic VOC, anthropogenic VOC, soil NOx, biomass burning CO etc.)

7. Line 285. "Strong vertical convection". Just curious, is there any index to show the strength the convection?

8. Line 340-342. In addition of NOx emission reduction, how about the changes of reaction rates and biogenic VOC emission caused by T reduction during pre-summer monsoon? And their roles in the O3 production?

9. Line 375. Can you show the selected region in one of your figures?

10. Line 384. How the statistics are conducted? How many samples are compared? Are the correlations for grid-to-grid? Or just the regional averages?

11. Figure 7. The caption is not clear. Do you mean " Differences in May-August mean between the lowest and highest SASMI conditions "?

12. Figure 3. Dry deposition is not shown here. As there is no obvious season changes? A little bit explanation would be helpful.

13. Figure 4. Can you give the number pairs of data?

---

## Referee Comment (RC2) · Anonymous Referee #2 · 14 Dec 2017

The study of Lu et al. analyzes the processes influencing lower tropospheric ozone over the Indian subcontinent with a focus on the influence of the South Asian monsoon but also analyzing interannual variability and trends. It mainly relies on a set of 20-year (1990-2010) long GEOS-CHEM simulations and analyzing the lower tropospheric ozone budget with respect to contributions from photochemistry, transport and deposition. The model-derived results are backed up with lower tropospheric ozone data derived from OMI satellite observations of the years 2006 - 2014 which, overall, show very good agreement with the model in terms of spatial distribution and seasonal behavior.

The publication is very well written (with small grammatical issues that will require some copy-editing), clearly structured, and the analyses are comprehensive and convincing.

In particular, the authors present a thorough analysis of the factors driving ozone variability over India including meteorological variability (seasonal, interannual), variations in anthropogenic and biomass burning emissions, and trends in methane. The paper has a good chance to become an important reference for future studies on the lower tropospheric ozone budget over India.

I support publication of this manuscript and have only a few small comments that may require minor revisions:

- As pointed out by Lelieveld et al. (The Indian Ocean Experiment: Widespread Air Pollution from South and Southeast Asia, Science, 2001), the ratio of emissions of NOx to those of CO and VOCs is much smaller over India than e.g. over Europe or the United States. As a consequence ozone production over India is likely strongly NOx-limited. It would have been nice if the authors had paid somewhat more attention to this aspect, notably in the analysis of the factors driving the long-term increase in ozone.

- Figure 2c shows the seasonal cycle of lower tropospheric ozone together with the different monthly production and loss terms. Why do these terms not sum up to positive values when the total burden of ozone increases? Take e.g. the budget for the month of March: The losses sum up to about -12 Tg, the gains only to about +11 Tg. Despite this negative budget, the ozone concentrations increase. from March to April. Please clarify.

- Page 4, line 86: decreases in ozone associated with the summer monsoon had also been reported based on MOZAIC aircraft observations, e.g. Srivastava et al. (Atmos. Env., 2015) or Bhattacharjee et al. (Meteorol. and Atmos. Phys., 2015).

- It was not always entirely clear, which regions the individual budgets and concentrations were representing, i.e., over which domain the values were averaged. "Lower troposphere over India" is not an exact term. Is this only over land, or over a rectangular domain covering India? I suggest adding such information to the figure captions.

- The same problem holds for some of the stated correlations, e.g. the correlation of r = 0.81-0.97 mentioned on page 10, line 223. What exactly was correlated here?

- Page 15, line 343: The statement "the mean correlation over India is still dominated by the temperature impact on ozone chemical production" is not quite correct. I would argue that high temperatures and high ozone levels are both a result of intense solar radiation, rather than temperature being directly the main driver for high ozone production.

- Page 16, Eq. 1: The grid indices i,j should probably be discarded, since equation 2 suggests that the norm is calculated by integration over a domain rather than a single grid cell. Probably the manuscript would also benefit from a schematic figure of the ozone budget in the pre-monsoon and monsoon seasons, i.e., a box representing the volume of the lower troposphere over India and the individual budget terms shown as arrows (in case of transport and deposition) or just numbers (in case of chemical production and loss within the box).

---

## Author Comment (AC1) · 25 Jan 2018

**Comment:** The manuscript by Lu et al. investigated the lower tropospheric ozone over India and its linkage to the south Asian Monsoon by using satellite observations and GEOS-Chem model. Spatial and temporal characteristics of lower tropospheric ozone over India were analyzed in terms of seasonal cycle and inter-annual variability (for 2006-2010) and also long term-trends (for 1990-2010). The contribution/roles of different processes, including precursor emissions (anthropogenic NOx, biomass burning etc), meteorology (horizontal and vertical transport), chemical production, and dry deposition were discussed. The linkage of lower tropospheric ozone concentrations over India to the south Asian Monsoon were also quantified.

[Figure]

This comprehensive analysis focus on the ozone over Indian region where the precursor emissions are still rising in recent years. This study is thorough and clear, which would help to understand the local and global environment effects of India ozone. Overall, this manuscript is well organized, states the problem, outlines the model experiments, and describes the model results. This study fits the scope of ACP. I recommend publication.

Below are several comments that I think the authors may address to improve the manuscript.
**Response:** We thank the reviewer for the valuable comments. All of them have been implemented in the revised manuscript. Please see our itemized responses below.

**Comment:** General comments:
1. Line 162-163. The authors reduced the 1990-1996 ACCMIP emissions by 30% to correct the gap between GFED3 and ACCMIP, and also to get a full set of data for 1990-2010. It's quite understandable and straightforward. While the readers might get the impression that GFED3 is more accurate than ACCMIP. I'm just curious is there any reference that evaluate the two biomass burning emission inventories for this region? Any discussions of the sensitivity of the biomass burning emissions would be helpful.
**Response:** Thanks for pointing it out; we do not have enough evidence from the literature to conclude which biomass burning emission inventory is more accurate over India. We now state in the text "Comparison of GFED3 and ACCMIP biomass burning CO emissions for their overlapping years (1997–2000) suggests ACCMIP is 30% higher. Here we reduce the 1990–1996 ACCMIP emissions by 30% to reconcile the two inventories, although this may lead to underestimates of biomass burning emission contributions for the period. We find that biomass burning emissions of CO over India (2.6 Tg a-1 (per annum) for 2006–2010) are relatively small compared with anthropogenic emissions (61.9 Tg a-1)".

[Figure]

**Comment:** 2. Line 177. The contributions of different processes are analyzed, including chemical production and loss, horizontal and vertical transport and dry deposition. I'm wondering if the cloud-chemistry/wet scavenging are considered as the two processes might involve the O3 chemistry. Or are they already considered but just ignored as the contribution are too small? Some discussion would be helpful.

**Response:** Cloud chemistry and wet deposition are included in the GEOS-Chem model, but they do not affect ozone concentrations directly. We now state here in the Section 2.3: "The GEOS-Chem model also include cloud chemistry (e.g., formation of sulfate aerosol via aqueous-phase reactions with ozone and H2O2) and wet deposition of soluble gases. The two processes have small effects on ozone directly due to its low solubility and thus are not diagnosed here".

**Comment:** 3. Line 181. For the horizontal transport (E-W, N-S) in the text and also figure 2, it would be better to give the directions for positive/negative values.

**Response:** We now state here in the text "Horizontal transport for each grid is calculated by horizontal fluxes from or to adjacent grids. Here we define transport from west to east or from south to north as positive values", and in the caption of Figure 2 "Horizontal transport from west (W) to east (E) and from south (S) to north (N), and downward (D) vertical transport at 600 hPa are defined as positive values".

**Comment:** 4. Section 3.1. The anthropogenic NOx emissions and biomass burning emissions are shown in this section while no discussions of NMVOC emissions are provided. While anthropogenic and biogenic NMVOC are also important ozone precursors. Especially for biogenic NMVOC, it might have strong seasonal cycle which might change the NOx/VOC regime in different seasons. Some discussions would be necessary here.

**Response:** We now show in Figure S1 the seasonal variations of anthropogenic NMVOC emissions, biogenic isoprene emissions, and soil NO emissions. This is also for addressing Comment 6 below. We now state in the text "Anthropogenic CO and

NMVOC emissions over India are 61.89 Tg a-1 and 15.5 Tg a-1, respectively, with similar seasonal variations as anthropogenic NO emissions (Figure S1)".

We also discuss in this section biogenic isoprene emissions and NOx/VOC regimes, "Model calculated biogenic isoprene emissions in India are 39.8 Tg C a-1, with a strong seasonality peaking in May and June (Figure S1). Previous studies have shown that the ratio of NOx emissions to CO and NMVOCs emissions over India is relatively small compared to other regions at northern mid-latitudes (Lelieveld et al., 2001; Li et al., 2017). Here we also examine the model simulated $H_2O_2/HNO_3$ concentration ratios, which have been used as an indicator of ozone production chemical regime (Sillman 1997; Zhang et al., 2016). We find that the $H_2O_2/HNO_3$ ratios in the Indian lower troposphere range from 1.0 to 5.0 for all four seasons, higher than those in eastern China and eastern US (Figure S2). This indicates strong NOx-limited conditions for ozone chemical production over India, consistent with previous studies (Kumar et al., 2012; Sharma et al., 2016)".

Added references:
Lelieveld, J., Crutzen, P. J., Ramanathan, V., Andreae, M. O., Brenninkmeijer, C. M., Campos, T., Cass, G. R., Dickerson, R. R., Fischer, H., de Gouw, J. A., Hansel, A., Jefferson, A., Kley, D., de Laat, A. T., Lal, S., Lawrence, M. G., Lobert, J. M., Mayol-Bracero, O. L., Mitra, A. P., Novakov, T., Oltmans, S. J., Prather, K. A., Reiner, T., Rodhe, H., Scheeren, H. A., Sikka, D., and Williams, J.: The Indian Ocean experiment: widespread air pollution from South and Southeast Asia, Science, 291, 1031-1036, 10.1126/science.1057103, 2001.
Li, M., Zhang, Q., Kurokawa, J.-I., Woo, J.-H., He, K., Lu, Z., Ohara, T., Song, Y., Streets, D. G., Carmichael, G. R., Cheng, Y., Hong, C., Huo, H., Jiang, X., Kang, S., Liu, F., Su, H., and Zheng, B.: MIX: a mosaic Asian anthropogenic emission inventory under the international collaboration framework of the MICS-Asia and HTAP, Atmos. Chem. Phys., 17, 935-963, 10.5194/acp-17-935-2017, 2017.
Sillman, S., He, D., Cardelino, C., and Imhoff, R. E.: The Use of Photochemical Indicators to Evaluate Ozone-NOx-Hydrocarbon Sensitivity: Case Studies from Atlanta, New York, and Los Angeles, J Air Waste Manag Assoc, 47, 1030-1040, 10.1080/10962247.1997.11877500, 1997.

Zhang, Y., Cooper, O. R., Gaudel, A., Thompson, A. M., Nédélec, P., Ogino, S.-Y., and West, J. J.: Tropospheric ozone change from 1980 to 2010 dominated by equatorward redistribution of emissions, Nature Geosci., 9, 875-879, 10.1038/ngeo2827, 2016.

**Comment:** 5. Line 213. When retrieving the OMI observed tropospheric ozone, some discussions of the sensitivity of the priori profile and average kernel matrices would be helpful. Are there any system biases or different biases in different seasons? Any uncertainties?

**Response:** Thanks for the suggestion. We now state in Section 2.1 (OMI satellite observations) "The degrees of freedom for signals (sum of the diagonal elements of averaging kernel matrices) for OMI ozone retrievals are typically 0.3–0.5 in the lower troposphere over India. Previous evaluations of the OMI retrievals with ozonesonde measurements have shown a clear improvement over the a priori in the lower troposphere of the tropics (30°S-30°N), and the mean retrieval biases in the tropics are less than 6% with little seasonality (Huang et al., 2017a)".

**Comment:** 6. Line 242. It's quite important to state that biogenic isoprene and soil NOx emissions are higher in the pre-summer monsoon than wintertime which help the reader to understand the results. As suggested in Q4, it will be great to give a table that lists the seasonal/annual emissions from different sources (biogenic VOC, anthropogenic VOC, soil NOx, biomass burning CO etc.)

**Response:** Thanks for the suggestion. We now add Figure S1 in the Supplement to show seasonal variations of anthropogenic NMVOCs, biogenic isoprene, and soil NO emissions, as described in Comment 4 above. We state in the text: "Biogenic isoprene emissions over India increases from 1.8 Tg month-1 in winter to 5.2 Tg C month-1 in

the pre-summer monsoon season. The soil NO emissions also increase from 0.08 Tg month-1 to 0.21 Tg month-1 (Figure S1)."

**Comment:** 7. Line 285. "Strong vertical convection". Just curious, is there any index to show the strength the convection?
**Response:** We now state in the text "Strong vertical convection with the 600 hPa upward velocity greater than 5 mm s-1 in July–August effectively uplifts ozone pollution from the lower troposphere to the upper troposphere".

**Comment:** 8. Line 340-342. In addition of NOx emission reduction, how about the changes of reaction rates and biogenic VOC emission caused by T reduction during presummer monsoon? And their roles in the O3 production?
**Response:** We have discussed in Section 3.2 (Variations in the pre-summer monsoon season) the influences of changes in temperature on ozone concentration through changing natural emissions and chemical production in the pre-summer monsoon. This sentence here was intended to emphasize the benefits of controlling NOx emissions. We now remove this sentence to avoid confusion.

**Comment:** 9. Line 375. Can you show the selected region in one of your figures?
**Response:** We now show the selected region for calculating the South Asian summer monsoon index in Figure S2. We state in the text "we then average $\delta(i, j)$ over the region of 35°E–90°E, 5°N–35°N (Figure S2) at 850 hPa and over May–August to represent the South Asian summer monsoon index (SASMI)."

**Comment:** 10. Line 384. How the statistics are conducted? How many samples are compared? Are the correlations for grid-to-grid? Or just the regional averages?
**Response:** We now state here "Interannual variations of Indian regional mean

lower tropospheric ozone concentrations are significantly negative correlated with the SASMI, as can be seen for both OMI observations (r = −0.46, 2006–2014, n = 9) and GEOS-Chem BASE results (r = −0.52, 1990–2010, n = 21)."

**Comment:** 11. Figure 7. The caption is not clear. Do you mean " Differences in May-August mean between the lowest and highest SASMI conditions "?
**Response:** Yes, we now clarify in the caption "Differences in May–August monthly mean (a) lower tropospheric ozone concentration, (b) . . . between the lowest and highest SASMI conditions. Values are calculated using averages of the five lowest SASMI years minus averages of the five highest SASMI years".

**Comment:** 12. Figure 3. Dry deposition is not shown here. As there is no obvious season changes? A little bit explanation would be helpful.
**Response:** We now present in Figure S1-d the spatial distribution and seasonal variation of ozone dry deposition to India. We also state in the text "Dry deposition of ozone to India shows a weak seasonal variation (1.5 ± 0.15 Tg month-1; Figure 2c and Figure S1-d)."

**Comment:** 13. Figure 4. Can you give the number pairs of data?
**Response:** We now state in the figure caption "Interannual correlation coefficients (r) between surface temperature and ozone (number of regional averages n = 9 for observations and 21 for model results) are shown inset".

---

## Author Comment (AC2) · 25 Jan 2018

**Comment:** The study of Lu et al. analyzes the processes influencing lower tropospheric ozone over the Indian subcontinent with a focus on the influence of the South Asian monsoon but also analyzing interannual variability and trends. It mainly relies on a set of 20-year (1990-2010) long GEOS-CHEM simulations and analyzing the lower tropospheric ozone budget with respect to contributions from photochemistry, transport and deposition. The model-derived results are backed up with lower tropospheric ozone data derived from OMI satellite observations of the years 2006 - 2014 which, overall, show very good agreement with the model in terms of spatial distribution and seasonal behavior.

[Figure]

The publication is very well written (with small grammatical issues that will require some copy-editing), clearly structured, and the analyses are comprehensive and convincing. In particular, the authors present a thorough analysis of the factors driving ozone variability over India including meteorological variability (seasonal, interannual), variations in anthropogenic and biomass burning emissions, and trends in methane. The paper has a good chance to become an important reference for future studies on the lower tropospheric ozone budget over India.

I support publication of this manuscript and have only a few small comments that may require minor revisions:
**Response:** We thank the reviewer for the valuable comments. All of them have been implemented in the revised manuscript. Please see our itemized responses below.

**Comment:** - As pointed out by Lelieveld et al. (The Indian Ocean Experiment: Widespread Air Pollution from South and Southeast Asia, Science, 2001), the ratio of emissions of NOx to those of CO and VOCs is much smaller over India than e.g. over Europe or the United States. As a consequence ozone production over India is likely strongly NOx-limited. It would have been nice if the authors had paid somewhat more attention to this aspect, notably in the analysis of the factors driving the long-term increase in ozone.
**Response:** Thanks for the suggestion. We now discuss the ratio of $H_2O_2/HNO_3$ as an indicator of ozone chemical production regime as presented in Figure S2. We state in Section 3.1 (Variations of meteorology and emissions): "Previous studies have shown that the ratio of NOx emissions to CO and NMVOCs emissions over India is relatively small compared to other regions at northern mid-latitudes (Lelieveld et al., 2001; Li et al., 2017). Here we also examine the model simulated $H_2O_2/HNO_3$ concentration ratios, which have been used as an indicator of ozone production chemical regime (Sillman 1997; Zhang et al., 2016). We find that the $H_2O_2/HNO_3$ ratios in the Indian lower troposphere range from 1.0 to 5.0 for all four seasons, higher than those in eastern China and eastern US (Figure S2). This indicates strong NOx-limited

conditions for ozone chemical production over India, consistent with previous studies (Kumar et al., 2012; Sharma et al., 2016)".

We also state in Section 5 (Long term trend), "Increases in anthropogenic NO emissions (about 3% year-1, Supplemental Figure S5) likely dominate the ozone increases due to the NOx-limited ozone production condition over this region as we discussed above".

Added references:
Kumar, R., Naja, M., Pfister, G. G., Barth, M. C., Wiedinmyer, C., and Brasseur, G. P.: Simulations over South Asia using the Weather Research and Forecasting model with Chemistry (WRF-Chem): chemistry evaluation and initial results, Geoscientific Model Development, 5, 619-648, 10.5194/gmd-5-619-2012, 2012.
Sharma, S., Chatani, S., Mahtta, R., Goel, A., and Kumar, A.: Sensitivity analysis of ground level ozone in India using WRF-CMAQ models, Atmos. Environ., 131, 29-40, 10.1016/j.atmosenv.2016.01.036, 2016.
Sillman, S., He, D., Cardelino, C., and Imhoff, R. E.: The Use of Photochemical Indicators to Evaluate Ozone-NOx-Hydrocarbon Sensitivity: Case Studies from Atlanta, New York, and Los Angeles, J Air Waste Manag Assoc, 47, 1030-1040, 10.1080/10962247.1997.11877500, 1997.
Zhang, Y., Cooper, O. R., Gaudel, A., Thompson, A. M., Nédélec, P., Ogino, S.-Y., and West, J. J.: Tropospheric ozone change from 1980 to 2010 dominated by equatorward redistribution of emissions, Nature Geosci., 9, 875-879, 10.1038/ngeo2827, 2016.

**Comment:** - Figure 2c shows the seasonal cycle of lower tropospheric ozone together with the different monthly production and loss terms. Why do these terms not sum up to positive values when the total burden of ozone increases? Take e.g. the budget for the month of March: The losses sum up to about -12 Tg, the gains only to about +11 Tg. Despite this negative budget, the ozone concentrations increase. from

March to April. Please clarify.

**Response:** Thanks for pointing it out. We find an error in calculation of vertical ozone fluxes for January to March. It is now corrected in the figure, and does not affect our conclusions.

**Comment:** Page 4, line 86: decreases in ozone associated with the summer monsoon had also been reported based on MOZAIC aircraft observations, e.g. Srivastava et al. (Atmos. Env., 2015) or Bhattacharjee et al. (Meteorol. and Atmos. Phys., 2015).

**Response:** These studies are now cited in the manuscript. We state here "Decreases of tropospheric ozone with the summer monsoon in South and East Asia have been reported from . . . aircrafts measurements (Bhattacharjee et al., 2015; Srivastava et al., 2015; Ojha et al., 2016) . . .".

Added references:

Bhattacharjee, P. S., Singh, R. P., and Nédélec, P.: Vertical profiles of carbon monoxide and ozone from MOZAIC aircraft over Delhi, India during 2003–2005, Meteorology and Atmospheric Physics, 127, 229-240, 10.1007/s00703-014-0349-x, 2014.

Srivastava, S., Naja, M., and Thouret, V.: Influences of regional pollution and long range transport over Hyderabad using ozone data from MOZAIC, Atmos. Environ., 117, 135-146, 10.1016/j.atmosenv.2015.06.037, 2015.

Ojha, N., Pozzer, A., Rauthe-Schöch, A., Baker, A. K., Yoon, J., Brenninkmeijer, C. A. M., and Lelieveld, J.: Ozone and carbon monoxide over India during the summer monsoon: regional emissions and transport, Atmos. Chem. Phys., 16, 3013-3032, 10.5194/acp-16-3013-2016, 2016.

**Comment:** - It was not always entirely clear, which regions the individual budgets and concentrations were representing, i.e., over which domain the values were averaged. "Lower troposphere over India" is not an exact term. Is this only over land,

or over a rectangular domain covering India? I suggest adding such information to the figure captions.

**Response:** We now state in the text "In this study, we present an integrated analysis of the processes controlling lower tropospheric (surface to 600 hPa) ozone concentrations over the terrestrial land of India and their linkage to the South Asian monsoon." We have also clarified it in the figure captions.

**Comment:** - The same problem holds for some of the stated correlations, e.g. the correlation of r = 0.81-0.97 mentioned on page 10, line 223. What exactly was correlated here?

**Response:** Thanks for pointing it out. We now state here "they generally capture the seasonal and spatial variations of OMI observed lower tropospheric ozone concentrations (r = 0.81–0.97 for the spatial variations of OMI observations vs. model results)." We have also clarified some other correlations in the text and figure captions.

**Comment:** - Page 15, line 343: The statement "the mean correlation over India is still dominated by the temperature impact on ozone chemical production" is not quite correct. I would argue that high temperatures and high ozone levels are both a result of intense solar radiation, rather than temperature being directly the main driver for high ozone production.

**Response:** We now clarify in the text: "the mean correlation over India still shows a positive effect of temperature on ozone chemical production, which can be driven by solar radiation affecting both temperature and ozone production rates, as well as the sensitivities of natural sources to temperature as discussed above."

**Comment:** - Page 16, Eq. 1: The grid indices i,j should probably be discarded, since equation 2 suggests that the norm is calculated by integration over a domain rather than a single grid cell.

**Response:** We rewrite some of the sentences here to avoid confusion. The monsoon

index is first calculated for each grid and then averaged over the South Asian region. We now state here "The monsoon index is first calculated for each model grid ($\delta$(i, j)) in the Northern Hemisphere in the month m and year y" and "we then average $\delta$(i, j) over the region of 35°E–90°E, 5°N–35°N (Figure S2) at 850 hPa and over May–August to represent the South Asian summer monsoon index (SASMI)".

**Comment:** Probably the manuscript would also benefit from a schematic figure of the ozone budget in the pre-monsoon and monsoon seasons, i.e., a box representing the volume of the lower troposphere over India and the individual budget terms shown as arrows (in case of transport and deposition) or just numbers (in case of chemical production and loss within the box).

**Response:** Thanks for the suggestion. We now present such a schematic figure in the main manuscript as Figure 4. We state in Section 3.3 (Variations in the summer monsoon season): "Figure 4 summarizes changes in the lower tropospheric ozone budgets over India in the pre-summer monsoon season (March-April) and the summer monsoon season (June-July-August). We can see that decreases in the Indian lower tropospheric ozone in the summer monsoon season are mainly associated with the reduction in ozone net chemical production and strengthening upward transport."